# *ARID1A*, NOTCH and WNT Signature in Gynaecological Tumours

**DOI:** 10.3390/ijms24065854

**Published:** 2023-03-19

**Authors:** Ieva Vaicekauskaitė, Daiva Dabkevičienė, Julija Šimienė, Diana Žilovič, Rūta Čiurlienė, Sonata Jarmalaitė, Rasa Sabaliauskaitė

**Affiliations:** 1National Cancer Institute, LT-08660 Vilnius, Lithuania; 2Institute of Biosciences, Life Sciences Center, Vilnius University, LT-08412 Vilnius, Lithuania

**Keywords:** ovarian cancer, NOTCH, WNT, ARID1A

## Abstract

Ovarian cancer (OC) is among the deadliest gynaecologic malignancies in the world. The majority of OC patients are diagnosed at an advanced stage, with high-grade serous OC (HGSOC). The lack of specific symptoms and suitable screening strategies lead to short progression-free survival times in HGSOC patients. The chromatin-remodelling, WNT and NOTCH pathways are some of the most dysregulated in OC; thus their gene mutations and expression profile could serve as diagnostic or prognostic OC biomarkers. Our pilot study investigated mRNA expression of the SWI/SNF chromatin-remodelling complex gene *ARID1A*, NOTCH receptors, WNT pathway genes *CTNNB1* and *FBXW7* mRNA expression in two OC cell cultures as well as 51 gynaecologic tumour tissues. A four-gene panel consisting of *ARID1A*, *CTNNB1*, *FBXW7* and *PPP2R1A* was used to investigate mutations in gynaecologic tumour tissue. All seven analysed genes were found to be significantly downregulated in OC when compared with non-malignant gynaecologic tumour tissues. *NOTCH3* was also downregulated in SKOV3 cells when compared to A2780. Fifteen mutations were found in 25.5% (13/51) of the tissue samples. *ARID1A* predicted mutations were the most prevalent with alterations detected in 19% (6/32) HGSOC and 67% (6/9) of other OC cases. Thus, *ARID1A* and NOTCH/WNT-pathway-related changes could be useful diagnostic biomarkers in OC.

## 1. Introduction

Ovarian cancer (OC) is the second leading cause of death from gynaecologic cancer after cervical cancer [1]. From the other gynaecologic diseases, OC stands out as particularly deadly, as the majority (48%) of cases are high-grade serous ovarian cancer (HGSOC) [2], a pathology often diagnosed in an advanced state (Grade 3, FIGO stage III or IV). The 5-year life expectancy after the diagnosis of HGSOC is only 26% for FIGO stage IV disease [3]. The rest of OC cases are highly heterogeneous with clear-cell, mucinous or endometrial histology, and even germ cell origin (granulosa tumours). The high rate of OC morbidity is attributed to the lack of specific symptoms and sufficient diagnostic techniques. Currently, the only OC biomarker serum CA125 is approved for HGSOC patient monitoring [4]. To compliment CA125, another serum biomarker, human epididymis protein 4 (HE4), has also been approved by the Food and Drug Administration (FDA) for endometrial OC follow-up; however, the biomarker is currently not recommended for clinical use due to inconsistent study results [4]. The OC screening efforts using serum biomarker and transvaginal ultrasound tests have shown no reduction in mortality rates [2]. Thus, OC patients urgently need new diagnostic and predictive biomarkers.

Recent studies have highlighted chromatin-remodeling factors as potential drivers and therapeutic targets of cancer. These complexes involve a substantial collection of proteins that epigenetically govern gene expression and are involved in reparation and replication [5]. *ARID1A*, coding a SWI/SNF complex protein, is the most-mutated gene in chromatin-remodeling complexes, with alterations detected in 6.2% of all solid tumours [6], and one of the most-mutated genes in OC. *ARID1A* mutations are mutually exclusive with *TP53* and are most indicative of endometriosis-linked OC types [7].

NOTCH and WNT/β-catenin are closely related pathways involved in female genital tract differentiation and cancerogenesis [8]. The two pathways converge at the primary NOTCH3 receptor ligand Jagged1, whose expression is regulated by the WNT/β-catenin pathway in OC [9]. The WNT signaling genes *CTNNB1*, *FBXW7* and *PPP2R1A* [10] are among the most-frequently mutated genes in gynaecologic malignancies [11], while changes in NOTCH family receptors and ligand expression are frequently detected in both malignant and non-malignant gynaecologic disorders [12].

Our pilot study investigated the expression of *ARID1A* and the NOTCH receptors and WNT components *CTNNB1* and *FBXW7* in two OC cell cultures, then validated the results in gynaecologic tumour tissues. In addition, a panel of four genes (*ARID1A*, *CTNNB1*, *FBXW7* and *PPP2R1A*) was analysed for predicted mutations in gynaecologic tumours.

## 2. Results

### 2.1. *ARID1A*, NOTCH/WNT Pathway Component mRNA Expression in Ovarian Cancer Cell Cultures

Out of the seven mRNAs analysed, all but *NOTCH4* expression were detectable in the two OC cell cultures. *NOTCH3* mRNA expression was significantly lower in the SKOV3 cell culture when compared with A2780 (−2.6 fold, *p* = 0.0002). *ARID1A* expression was also decreased in SKOV3, albeit not significantly (Figure 1).

### 2.2. *ARID1A*, NOTCH/WNT Pathway Component mRNA Expression in Gynaecological Cancer Tissues

The mRNA expression of all seven analysed genes was significantly downregulated in HGSOC tissues when compared to benign gynaecologic disease tissues (Figure 2A). The *NOTCH4* alongside *CTNNB1* and *FBXW7* mRNA was the most severely altered. However, when compared with non-HGSOC gynaecologic cancer cases, only WNT component downregulation proved significant in HGSOC tissues (Figure 2B).

Next, we evaluated the correlations between mRNA expression and clinical/pathological features in HGSOC samples. In the HGSOC tissue cohort, the *CTNNB1* mRNA was significantly downregulated in FIGO stage IV cases when compared with FIGO stage II or III OC (*p* = 0.02), while the *FBXW7* change was of borderline significance (*p* = 0.05) (Figure 3A). Moreover, *NOTCH1* and *NOTCH2* expression was significantly reduced in cases with residual tumours (R1) (*p* = 0.03 and *p* = 0.02, respectively) (Figure 3B). No correlation between mRNA expression in gynaecologic cancer tissues and age or pre-surgery serum CA125 concentration was found.

*ARID1A* expression showed an association of borderline significance (*p* = 0.049) with PFS: OC patients with higher-than-median *ARID1A* expression showed better PFS (36 vs. 32 months), while other gene expression had no association with PFS.

Overall, all seven mRNAs were good separators of gynaecologic cancer vs. benign disease (Figure 4A), with the *CTNNB1* showing the best diagnostic power (area under the curve (AUC) = 0.93). Multiple logistic regression of all seven mRNAs perfectly diagnosed gynaecologic cancer cases from the benign gynaecologic disease (AUC = 1). However, the mRNA expression was less accurate in separating HGSOC cases from other gynaecologic cancers (Figure 4B), with both WNT genes remaining the best separators. The combination of all seven mRNAs showed an acceptable level of HGSOC separation from the other gynaecologic cancers (AUC = 0.85).

### 2.3. *ARID1A* and WNT Pathway Gene Mutations in Gynaecologic Cancer Tissues

The cohort of 51 tissue samples was also investigated for mutations in *ARID1A*, *CTNNB1*, *FBXW7* and *PPP2R1A* mutations using targeted NGS. In all, 15 mutations were found in 25.5% (13/51) of the tissue samples (Figure 5). *ARID1A* alterations were the most prevalent, with 23.5% (12/51) of patients carrying predicted mutations. Only two alterations detected in *CTNNB1* and one in *PPP2R1A*. No mutations were detected in *FBXW7*. In both cases *CTNNB1* co-occurred with *ARID1A* predicted mutations.

Both *CTNNB1* alterations were exclusively detected in non-serous type cancer: one in clear-cell OC and the other in a case with simultaneous endometrioid ovarian and endometrial cancer.

*ARID1A* alterations were detected in 19% (6/32) of HGSOC and and 67% (6/9) of other OC cases; however, no tissue alterations were found in non-malignant tumours or the sole endometrial cancer case. *ARID1A* alterations were able to separate OC from benign gynaecologic tumours with 29% sensitivity and 100% specificity. No correlation between the predicted mutation status and the clinical data was found.

Although almost all mRNA expression was lower in cases with *ARID1A* predicted mutations, no statistically significant correlation between *ARID1A* alterations and gene expression was found.

None of the detected *ARID1A* alterations were listed in ClinVar as pathogenic mutations (no information or variants of uncertain significance (VUS)); however, only one *CTNNB1* was VUS, and the mutation in *PPP2R1A* was pathogenic (Table 1). Analysing the VUS alterations using public databases showed that 50% (5/10) of *ARID1A* alterations are likely to be pathogenic. All but one of these are truncating alterations. All but one ARID1A missense alterations were also predicted “damaging” by the in silico analysis.

## 3. Discussion

HGSOC is one of the leading causes of female cancer death in the world; thus, means of diagnosis and screening for this gynaecologic cancers are in urgent need. Our pilot study has shown the potential of NOTCH/WNT pathway components as well as the chromatin-remodelling complex member *ARID1A* as possible biomarkers of OC through an analysis of mRNA transcription and DNA mutations.

First, our small-scale cell culture mRNA expression analysis showed differences in gene expression between a more aggressive type OC cell line (SKOV3), often characterised as clear-cell or serous-type OC (based on *TP53* mutation) and a more endometrioid-type OC-representing cell line (A2780) [13]. The cell line analysis provided additional diversity in the types of OC analysed when compared with our tissue cohort. The two OC cell line mRNA expression comparison found differences in all analysed mRNAs; however, only *NOTCH3* expression was significantly lower in SKOV3 when compared to A2780. Conversely, Wang et al.’s study found a higher expression of NOTCH components (*NOTCH1* and *HES1*) in A2780 when compared with SKOV3 and three other ovarian cell cultures. The downregulation of NOTCH1 by γ-secretase inhibitors negatively affected cell growth and induced apoptosis [14]. In line with our study, NOTCH3 protein expression was found to be absent from SKOV3, while A2780 expressed NOTCH3 [15,16]. *FBXW7* expression was also found to be lower in SKOV3 when compared to A2780 [17]. Interestingly, both cell cultures have *ARID1A* mutations; however, SKOV3 also has *FBXW7* and *NOTCH2* mutations [13]. Due to *ARID1A* nonsense mutations, both OC cell cultures are deficient in full-length *ARID1A* expression [18]. To our best knowledge no other comparison of *ARID1A* or *CTNNB1* mRNA expression in the two cell cultures has been previously made.

Many studies including data available in the TCGA database find *NOTCH1–4*, especially *NOTCH3*, amplifications in a significant percent of HGSOC patients and high *NOTCH3* and other NOTCH component upregulation is associated with poor survival [8,16]. Our study showed downregulation of all genes tested. Downregulated NOTCH receptor expression in HGSOC could be related to our control group: in ectopic endometriosis, adenomyosis and other benign gynaecologic diseases, NOTCH receptor expression is increased [12]. We used a mix of various benign gynaecologic conditions (ovarian endometriosis, cystadenomas, myomas) as our control group; thus, the upregulation in HGSOC cases simply could have been less than in benign gynaecologic cancers.

Similarly to our study, significant NOTCH/WNT ubiquitin ligase gene *FBXW7* downregulation in serous OC samples was also reflected in Kitade et al.’s study. This study also found associations between lower *FBXW7* expression and more advanced OC stages, albeit not clinically significant. The proposed mechanism of this downregulation is hypermethylation of *FBXW7* 5’-upstream regions, which is related to the high prevalence of *TP53* mutations in HGSOC samples causing DNA methyltransferase 1 (DNMT1) overexpression [19].

*NOTCH1-2*’s downregulated expression was also linked with residual tumour after surgery. In other malignancies, such as gastric cancer and breast cancers, NOTCH receptor expression is a good prognostic indicator of residual disease, where low *NOTCH2* expression in residual tumours is correlated with longer survival times [20]. Conversely, in breast cancer, NOTCH signalling is active in residual tumour cells and is related to tumour recurrence [21].

Mutations in *CTNNB1* are found in around 43% of endometrioid ovarian carcinomas leading to a loss of β-catenin expression in 51% of cases [22]. Similarly, 18% of endometrioid endometrial carcinomas also possess *CTNNB1* mutations [23]. Typically, these mutations are missense alterations affecting the amino-terminal domain required for phosphorylation by GSK3β to signal degradation of the β-catenin. The activating *CTNNB1* mutations render the WNT pathway constantly active [24]. In our study predicted mutations in *CTNNB1* were found two types of OC tissues (a case with simultaneous ovarian and endometrial cancers and a clear-cell OC case). However, in OC tissues *CTNNB1* expression was the most significantly reduced when compared with benign gynaecologic malignancies and correlated with FIGO stage. The low expression of *CTNNB1* in FIGO grade IV HGSOC cases is particularly interesting as there were no associations between β-catenin protein expression and any of the clinical/pathological features of serous OC [25]. More studies are greatly needed in order to further validate the significance of *CTNNB1* downregulation in OC.

Protein phosphatase 2A gene *PPP2R1A* mutation P179R is enriched in high-grade endometrial carcinoma [26]; however, in our study it was found in HGSOC case. The mutation significantly reduces the stability, enzymatic activity and ligand binding to the PP2A catalytic subunit which leads to reduced dephosphorylation of GSK3β and β-catenin [26].

The *ARID1A* alterations in our study were found in tissue samples from virtually every type of OC; however, no alterations were detected in benign gynaecologic conditions. Typically, *ARID1A* loss is indicative of endometrioid-origin cancers; *ARID1A* mutations are found in 40% of endometrial cancers [27], 32% of endometrioid OCs, 29% of clear-cell OCs, and only 3% of HGSOC cases [28]. The De Leo et al. study, which examined *ARID1A* mutation’s effect on transcriptomic and proteomic levels of *ARID1A*, in line with our study results, found no significant correlation between *ARID1A* mutations and mRNA expression [27]. Although both *ARID1A* mutations and mRNA downregulation lacked specificity in HGSOC diagnosis, we found borderline significant associations of low *ARID1A* expression and reduced progression-free survival, showing the potential of *ARID1A* as a prognostic factor in OC. *ARID1A*’s loss is associated with PFS as *ARID1A* downregulation and mutations are highly associated with chemoresistance to platinum-based therapies [29,30]. Synthetic lethality strategies, such as DNA damage response and epigenetic regulation pathway-targeting drugs, should be explored with *ARID1A*-deficient OC [5].

*ARID1A* mutations alone are not substantial enough to cause malignancies by themselves [31]; thus, *ARID1A* mutations are often concomitant with mutations in other genes. In two cases, additional alterations in *CTNNB1* were found together with *ARID1A* mutations. Although none of the *ARID1A* mutations from our study was annotated in ClinVar (as of January 2023), 40% of them were truncating alterations likely to affect the viability of the ARID1A protein. However, even the non-truncating mutations in *ARID1A* could still affect the acetylation of histone tails on the nucleosomes and affect target gene expression epigenetically, as ARID1A is a core component of the SWI/SNF chromatin-remodelling complex [32]. More functional studies are greatly needed in order to determine *ARID1A* mutation pathogenicity and possible implications for OC development.

Due to the pilot study design our analysis was limited by the available tissue samples and OC cell cultures. The small and heterogenic sample cohort limited the statistical power to test associations among mutation, mRNA expression and the clinical data. A larger number of samples should be tested to validate and expand on our results.

## 4. Materials and Methods

### 4.1. Cell Culture

Two ovarian cell cultures, SKOV3 and A2780, were tested for the seven genes’ (*NOTCH1–4*, *ARID1A*, *CTNNB1* and *FBXW7*) mRNA expression. Both cell cultures are the two best-characterized and most-used OC cell cultures; although they do not closely resemble HGSOC, they were chosen to represent endometrioid OC (A2780) and clear-cell OC (SKOV3), the latter regarded as a more aggressive type of OC (although SKOV3 is frequently assumed to represent serous OC due to mutation in *TP53*) [33].

Ovarian cancer cells SKOV3 and A2780 were kindly provided by the NCI Immunology laboratory. A2780 cells were cultured in RPMI 1640 + GlutaMax™-I (Gibco, TFS, Grand Island, NY, USA) medium, while SKOV3 cells were cultured in DMEM + GlutaMax™-I (Gibco, TFS, Grand Island, NY, USA) medium. Both cell media were supplemented with 10% fetal bovine serum (Gibco, TFS, Grand Island, NY, USA) and penicillin/streptomycin (Invitrogen Life Technologies, Carlsbad, CA, USA). Cells were cultivated in a monolayer in 37 °C, 5% CO${}_{2}$ and humidified atmosphere conditions for 48 h until RNA extraction.

### 4.2. Patient Cohort and Sample Collection

The tissue study cohort consisted of 51 patients who underwent salpingoovarectomy at the Lithuanian National Cancer Institute between 2018 and 2021 for the removal of ovarian or endometrial tumours or benign gynaecologic tumours, including one patient who underwent prophylactic salpingoovarectomy because of a germline *BRCA2* mutation (risk-reducing surgery (RSS)). The 51 patients were divided into three groups: HGSOC (32 cases); other gynaecologic tumours (10 cases), which included 9 cases of non-HGSOC ovarian tumours and one case of endometrial cancer. The third group, regarded as controls (9 cases), was made up of 8 benign gynaecologic tumours and one prophylactic salpingoovarectomy case. The clinical features of the tissue sample cohorts are in the Table 2.

During salpingoovarectomy, a small part of the tumour samples were allocated for the study and stored immediately at −80 °C until nucleic acid extraction. The study was approved by the regional bioethics committee (No. 158200-18/5-988-539). All patients were informed about the study and signed written informed consent forms.

### 4.3. Nucleic Acid Extraction

Prior to nucleic acid extraction, tissue samples were homogenized in liquid nitrogen using a mortar and pestle. 10–15 mg of the resulting tissue was used for nucleic acid extraction. Total RNA extraction for tissue and cell culture samples was performed using TRIzol reagent (Invitrogen, TFS, Carlsbad, CA, USA) using a standard protocol, while for the DNA extraction, tissues first underwent a 16-hour digestion with proteinase K solution (ThermoScientific, TFS, Vilnius, Lithuania) and then standard phenol–chloroform extraction and ethanol precipitation protocols were applied. The nucleic acid samples were stored at −80 °C until further use. The RNA quantity and quality was evaluated using a Nanodrop 2000 spectrophotometer (Thermo Scientific, Wilmingron, DE, USA), while DNA quantification was conducted with a Qubit™ dsDNA HS Assay Kit on a Qubit™ 2.0 Fluorimeter (Invitrogen, TFS, Eugene, OR, USA) according to the manufacturer’s instructions.

### 4.4. cDNA Synthesis and Quantitative PCR

The RNA samples from ovarian cancer cell cultures and tissue samples were used for the quantitative analysis of NOTCH receptors (*NOTCH1-4*), *FBXW7* and the β-catenin gene *CTNNB1*, as well as the chromatin-remodelling complex SWI/SNF subunit coding gene *ARID1A* mRNA transcripts. First, cDNA was synthesised from the total RNA samples using the Maxima First Strand cDNA Synthesis Kit for RT–qPCR with dsDNase (ThermoScientific, TFS, Vilnius, Lithuania) on a ProFlex PCR System (Applied Biosystems, TFS, Singapore). The resulting cDNA was used for quantitative PCR (qPCR) using a Maxima SYBR Green qPCR Master Mix (2X) kit (ThermoScientific, TFS, Vilnius, Lithuania) on a QuantStudio 5 Real-Time PCR System (Applied Biosystems, TFS, Singapore). The primer sequences are provided in Appendix A Table A1. All qPCR reactions were performed in duplicate according to the manufacturer’s protocol. The initial Ct values were gathered using QuantStudio Design & Analysis Software v1.4.3 (Applied Biosystems) with automatic baseline. Then, the data were normalized to a reference gene (*GAPDH*) and log_2_ 2^−ΔCt^ values and used for further statistical analysis.

### 4.5. Targeted Next-Generation Sequencing

The 51 tissue samples were analysed for mutations in the *ARID1A*, *CTNNB1*, *FBXW7* and *PPP2R1A* genes using targeted next-generation sequencing (NGS). The library preparation was carried out according to the manufacturer’s protocol using an Ion AmpliSeq™ Library Kit 2.0 and a custom On-Demand Panel (Life Technologies (LT), Carlsbad, CA, USA). Quantification of the final libraries was performed using an Ion Library TaqMan™ Quantification Kit (AB, TFS, Vilnius, Lithuania). An equal amount of each sample library was used for sequencing with the Ion Torrent™ Ion S5™ system. Sequencing data analysis was performed using the Ion Reporter 5.18 tool (LT, Carlsbad, CA, USA). First, the sequence reads were aligned to human reference genome 19 (Genome Reference Consortium GRCh38); then, each alignment was additionally visualized and verified on the Integrative Genomics Viewer 2.4.8 tool. Each mutation was classified according to the ClinVar [34] database as pathogenic or likely pathogenic, benign or likely benign, and mutations with conflicting evidence of pathogenicity and variants not yet included in ClinVar as of January 2023 were classified as variant uncertain significance (VUS). Only pathogenic, likely pathogenic and VUS were included in the analysis. The VUS alterations were predicted pathogenic mutations according to the analysis performed using the Varsome (varsome.com, accessed on January 2023 [35]) and Varcards (varcards.biols.ac.cn, accessed on January 2023 [36]) databases.

### 4.6. Statistical Analysis

Data normality was tested using the Shapiro–Wilk W test. As appropriate, associations between categorical data variables were determined using a two-sided Chi-square test or Fisher’s exact test, while associations between two independent samples were analysed with a Mann–Whitney U test or Welch’s t test. Receiver operator curves were applied to determine the biomarker sensitivity and specificity. Logistic regression probabilities were used for combining multiple biomarkers into a singular test model. Kaplan–Meier curve analysis were applied for progression-free survival (PFS) analyses. PFS at one year was denoted as the time from treatment (surgery) to progression or death. Results were regarded as statistically significant when *p* value ≤ 0.050. All statistical data analysis and visualisation was conducted using R x64 4.0.3, GraphPad Prism 8 and MedCalc 14.8.1 softwares.

## 5. Conclusions

Our pilot study reveals a significant deregulation of NOTCH receptor expression, as well as expression changes and mutations in *ARID1A* and the WNT pathway genes *CTNNB1* and *FBXW7* in gynaecologic tumours. Alterations in the chromatin-remodeling and NOTCH/WNT pathways in the future could serve as novel diagnostic or prognostic biomarkers for gynaecologic malignancies. More expansive studies are required for tissue biomarker validation in non-invasive liquid biopsy samples as well as the clinical use of genetic biomerkers for OC detection.

## Figures and Tables

**Figure 1 ijms-24-05854-f001:**
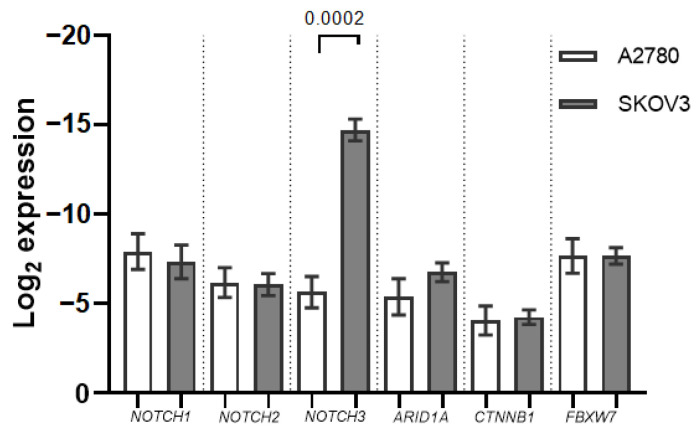
mRNA expression in ovarian cancer (OC) cell cultures A2780 and SKOV3. Boxes indicate the mRNA expression normalized to *GAPDH* expression data. Whiskers denote standard deviation from the three experimental replicates.

**Figure 2 ijms-24-05854-f002:**
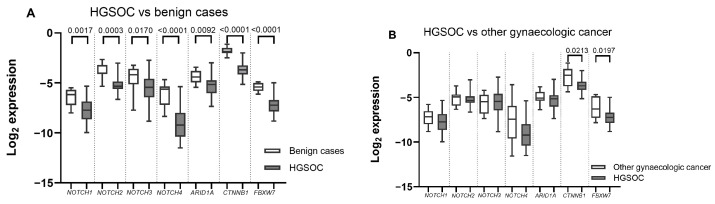
mRNA levels in ovarian cancer tissues. (**A**) gynaecologic cancer (n = 42) vs. benign cases (n = 9) (**B**) high-grade serous ovarian cancer (HGSOC) (n = 32) vs. other gynaecologic cancer cases (n = 10). Boxes indicate the log${}_{2}$ normalized expression data. Whiskers denote minimum and maximum values.

**Figure 3 ijms-24-05854-f003:**
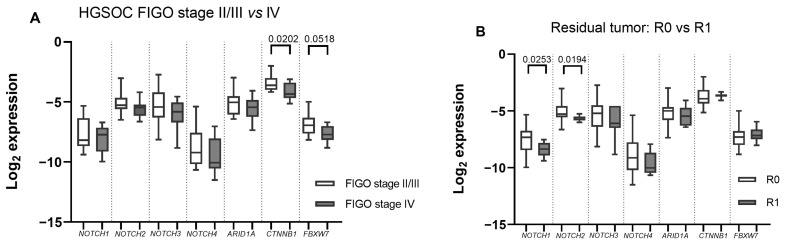
mRNA levels in high-grade serous ovarian cancer HGSOC tissues: (**A**) FIGO stage II/III (n = 20) vs. IV (n = 12); (**B**) with (R1) (n = 7) vs. without (R0) residual tumour (n = 25). Boxes indicate the log${}_{2}$ normalized expression data. Whiskers denote minimum and maximum values.

**Figure 4 ijms-24-05854-f004:**
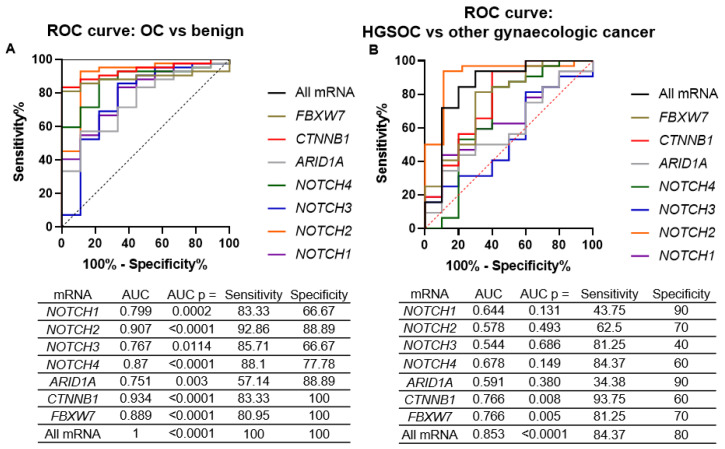
ROC curve analysis of tissue mRNA biomarkers. (**A**) gynaecologic cancer (ovarian cancers + one endometrial case) vs. benign cases. (**B**) HGSOC cases vs. other gynaecologic cancer cases. AUC—area under the curve.

**Figure 5 ijms-24-05854-f005:**
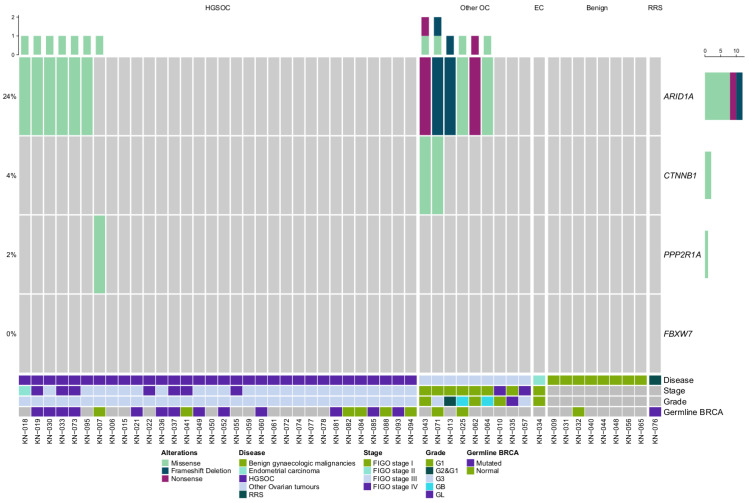
Oncoprint of gynaecologic tumour tissue sample mutation data. HGSOC—High-grade ovarian cancer, OC—ovarian cancer, EC—endometrial cancer, RRS—risk-reducing surgery.

**Table 1 ijms-24-05854-t001:** Observed alterations and their predicted pathogenicity.

No.	Samples	Histology/Disease Type	Locus	Gene	Coding Sequence	Amino Acid Change	Variant Effect	ClinVar	dbSNP	VarSome Verdict	VarCards “Damaging Score”
1	KN-007	HGSOC	chr19:52715971	*PPP2R1A*	c.536C>G	p.Pro179Arg	missense	Likely pathogenic	rs786205228	Likely pathogenic	0.83
2	KN-043	OC and EC	chr3:41266104	*CTNNB1*	c.101G>A	p.Gly34Glu	missense	Pathogenic/ Likely pathogenic	rs28931589	Pathogenic	0.83
3	KN-043	OC and EC	chr1:27106364	*ARID1A*	c.5975C>A	p.Ser1992Ter	nonsense			Pathogenic	1
4	KN-071	Clear-cell OC	chr3:41266113	*CTNNB1*	c.110C>G	p.Ser37Cys	missense	Pathogenic/ Likely pathogenic	rs121913403	Pathogenic	0.83
5	KN-071	Clear-cell OC	chr1:27057961	*ARID1A*	c.1670_1674delAGTCT	p.Gln557ProfsTer64	frameshift deletion			Likely pathogenic	-
6	KN-013	OC and EC	chr1:27099367	*ARID1A*	c.3606delG	p.Asn1203IlefsTer3	frameshift deletion			Pathogenic	-
7	KN-018	HGSOC	chr1:27099478	*ARID1A*	c.3715G>A	p.Ala1239Thr	missense			Likely pathogenic	0.64
8	KN-073	HGSOC	chr1:27023961	*ARID1A*	c.1067G>A	p.Arg356Lys	missense			Likely benign	0.52
9	KN-019	HGSOC	chr1:27106228	*ARID1A*	c.5839C>G	p.Gln1947Glu	missense			Likely benign	0.3
10	KN-033	HGSOC	chr1:27101532	*ARID1A*	c.4814C>T	p.Pro1605Leu	missense		rs375431469	VUS	0.61
11	KN-064	Serous borderline									
12	KN-025	Mucinous borderline	chr1:27089762	*ARID1A*	c.2718C>G	p.Asn906Lys	missense	Uncertain significance	rs201864573	Benign	0.65
13	KN-095	HGSOC									
14	KN-062	Mucinous borderline	chr1:27101504	*ARID1A*	c.4786G>T	p.Glu1596Ter	nonsense			Pathogenic	1
15	KN-030	HGSOC	chr1:27105686	*ARID1A*	c.5297A>T	p.Glu1766Val	missense		rs1363371199	Likely benign	0.39

**Table 2 ijms-24-05854-t002:** Clinical features of the patients with tissue samples.

Clinical/Pathological Characteristics	Number of Patients (%)			
**Disease Group**	**HGSOC**	**Other Gynaecologic Cancers**	**Benign Gynaecologic Tumour**	**Overall**
n =	32	10	9	51
Average Age, years (min–max)	57.8 (41–82)	63.7(49–77)	53.6 (43–72)	58.3 (41–82)
Average CA125 pre op. concentration U/mL (N/A)	888.5 (1 N/A)	139.4 (3 N/A)	51.35 (2 N/A)	641.8 (6 N/A)
FIGO Stage				
IA		8 (80.0)		8 (15.6)
IIB	1 (3.1)			1 (2.0)
IIIB	2 (6.3)			2 (4.0)
IIIC	17 (53.1)			17 (33.3)
IVB	12 (37.5)	2 (20.0)		14 (27.5)
N/A ^1^			9 (100.0)	9 (17.6)
Tumour differentiation grade				
G1		4 (40.0)		4 (7.8)
G2		1 (10.0)		1 (2.0)
G3	32 (100.0)	2 (20.0)		4 (66.7)
GB/GL ^2^		3 (30.0)		3 (5.9)
N/A1			9 (100.0)	9 (17.6)
Progressed disease				
Yes	12 (37.5)	1 (10.0)		13 (25.5)
No	20 (51.4)	9 (90.0)	9 (100.0)	38 (74.5)
Radical disease after surgery				
R0	24 (47.1)	8 (80.0)	9 (100.0)	41 (80.3)
R1	6 (18.8)	2 (20.0)		8 (15.7)
R3	1 (3.1)			1 (2.0)
N/A ^3^	1 (3.1)			1 (2.0)

^1^ N/A—benign cases ^2^ GB—borderline cases, GL—granulosa tumour ^3^ N/A—needle biopsy.

## Data Availability

Not applicable.

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
