# Peer review of "ARID1A*, NOTCH and WNT Signature in Gynaecological Tumours"

_ijms, 2023, doi:10.3390/ijms24065854_

Round 1
Reviewer 1 Report
The authors investigated the characteristic genes of NOTCH/WNT signaling pathways in ovarian cancer cell lines and patient tumors. By comparing HGSOC to benign tumors/other malignant gynecologic masses, they revealed that deregulation of NOTCH, ARID1A, CTNNB1 and FBXW7 have potential diagnostic values.
Major issues:
1. In the introduction part, diagnostic markers used for ovarian cancer should be delineated.
2. In targetd next generation sequencing workflow, GRCh38 should be used as a reference genome instead of older version of it. Sometimes, some people will still use older version because of some weired requirement for specific softwares. But justifying using older one should be delineated in methodology part.
3. There are different stages of OC in tumors. It will be better if you can derive conclusions from different comparisons.
Minor issues:
The legend for Figure 5 is too small to be legible. It will be better if you increase the font of it a little bit.
Reviewer 2 Report
Title: ARID1A, NOTCH and WNT signature in gynaecological tumours
In this manuscript, Vaicekauskaite et al. studied the feasibility of using expression levels in a small panel of cancer-related genes as diagnostic markers to detect ovarian cancer (OC). To do so, they focused on members of the NOTCH/WNT pathway as well as the SWI/SNF-subunit ARID1A. Mutations in these genes have been extensively described before. Therefore, the authors focused primarily on how this affects the gene expression in different OC subtypes as well as at different stages of the disease.
As the authors openly expressed the current OC cell models do not fully recapitulate the alterations found in tumor samples. In agreement to that, the cell line data presented here is less compelling and would probably require a more diverse cell line panel (as mentioned in the comments below). Despite that, the authors validated their initial findings in a large panel of patient samples, which provided a more comprehensive characterization of how these pathways are deregulated in OC.
The work described in this manuscript seems adequate for the scope of the journal and it is of interest for the cancer community. I would like to suggest a few changes that could strengthen this manuscript:
Line 6: It’s confusing when the authors refer to “predicted mutations”. According to the methods section (2.5), the entire gene regions were sequenced. Is that correct or only certain hotspots were sequenced? The authors should clarify it in the methods section and/or remove the concept of “predicted mutations”, if they were not aware about the presence of certain mutations prior to carry out targeted sequencing.
Line 56: “Cell lines” is preferable to “cell cultures”.
Line 143: mRNR?
Fig. 2 and 3: To help interpretation, the authors should indicate the number of samples in each group either in the text or the figure legend.
Line 154-159: The authors may consider introducing why they compare expression levels at different stages and with/without residual tumors after surgery.
Line 178: Two alterations co-occur or co-exist but they are not defined as “mutually inclusive”, in contrast to mutually exclusive mutations.
Line 201: Only one gene (NOTCH3) shows differential expression between the two cell lines. The authors can’t claim that gene expression is deregulated in SKOV3 compared to A2780. If the authors would like to go in this direction, I would expect some additional cell lines to compare mRNA expression in clear-cell type OC vs. endometrioid type OC.
Line 205-207: That’s not shown in figure 1. There is not statistical comparison allowing the authors to make such a claim.
Line 207: The authors probably mean higher expression instead of significant upregulation.
Line 211: The authors contradict themselves with what they claimed in lines 205-207 (see previous comment).
Comment: NOTCH is misspelled 9 times over the course of the manuscript. Please correct it.
